# Semi-supervised Active Learning for Left Ventricle Segmentation in Echocardiography

**Eman Alajrami**[1]                                                                    EMAN.ALAJRAMI@UWL.AC.UK

**Nasim DadashiSerej**[1]                                              NASIM.DADASHISEREJ@UWL.AC.UK

**Jevgeni Jevsikov**[1,2]                                                     J.JEVSIKOV@IMPERIAL.AC.UK

**Patricia Fernandes**[1]                                         PATRICIA.FERNANDES@UWL.AC.UK

**Abas Abdi** [1]                                                                           ABAS.ABDI@UWL.AC.UK

**Isreal Ufumaka** [1]                                                         ISREAL.UFUMAKA@UWL.AC.UK

**Darrel P Francis**[2]                                                                  DARREL@DRFRANCIS.ORG

**Massoud Zolgharni**[1,2]                                                   MASSOUD@ZOLGHARNI.COM

[1] *School of Computing and Engineering, University of West London, London, United Kingdom*

[2] *National Heart and Lung Institute, Imperial College, London, United Kingdom*

**Editors:** Under Review for MIDL 2024

## Abstract

Training deep learning models requires large labelled datasets, which are expensive and scarce in medical imaging. This study investigates semi-supervised active learning for left ventricle segmentation in echocardiography, aiming to reduce the need for extensive manual expert annotations. A novel technique for identifying reliable pseudo-labels is proposed. Results show a significant reduction in annotation efforts by up to 93%, achieving 99% of the maximum accuracy using only 7% of labelled data. The study contributes to efficient annotation strategies in medical image segmentation.

**Keywords:** Echocardiography, Deep learning, Semi-supervised active learning

## 1. Introduction

Left ventricle (LV) segmentation is vital for accurately assessing clinical parameters like LV ejection fraction (Azarmehr et al., 2020; Cole et al., 2015). Deep learning (DL) models, such as U-Net, have gained popularity in medical image analysis (Ronneberger et al., 2015). However, these models require large annotated datasets for training, posing challenges in medical imaging where annotations are scarce and costly. Active Learning (AL) and Semi-supervised Learning (SSL) have emerged as solutions to annotation challenges (Alajrami et al., 2024; Chen et al., 2021). Semi-supervised active learning (SSAL) combines AL and SSL to maximise the utilisation of unlabelled data. One popular approach in SSAL is the Cost-Effective Active Learning (CEAL), which uses pseudo-labels for confident model predictions and combines them with expert annotations of the most uncertain samples (Wang et al., 2016). However, CEAL lacks robust evaluation of pseudo-label quality, potentially impairing model training. We introduce a new approach for SSAL to address these challenges. Our method determines a range of candidate pseudo-labels from the samples of the highest uncertainty frequency in the uncertainty histogram. Predictions in this range will be sorted based on confidence scores to select reliable pseudo labels for training, combined with expert annotations of the uncertain images. It incorporates post-processing for pseudo-labels, refining them before adoption for training. We aim to evaluate the efficacy of the proposed approach in LV segmentation, rarely studied in this field, compared to existing methods, filling a gap in the literature.

Alajrami  DadashiSerej Jevsikov[1,2]  Fernandes    Francis Zolgharni

## 2. Method and dataset

**Development dataset:** Unity dataset contains 1224 videos of apical 4-chamber echocardiographic view, obtained from Imperial College Healthcare NHS Trust's database, captured between 2015 and 2016. The images were acquired following standard protocols by experienced echocardiographers using ultrasound equipment from GE and Philips. Ethical approval was granted by the Health Regulatory Agency. From these videos, 2800 images were sampled at various points in the cardiac cycle and labelled by individual experts using our in-house online platform (https://unityimaging.net). This dataset was utilised for model training and validation.

**Testing dataset:** This dataset included 100 videos collected over three consecutive working days in 2019, from which 200 end-diastolic and end-systolic frames were selected and labelled by 11 experts. The consensus of experts' annotations was used as ground truth in the testing dataset.

U-Net model of depth 5 was implemented in TensorFlow and trained on an Nvidia RTX3090 GPU for 100 epochs with early stopping and patience at 10.

We compared our method to Random, AL, and CEAL using Entropy for uncertainty scoring (Alajrami et al., 2024). Each method was evaluated on the testing dataset using Dice Coefficient (DC) (Naidoo et al., 2022).

From the training dataset, 4% (82 images) were selected for the initial labelled set $L$; the rest was considered as the unlabelled pool $U$. For AL methods, the batch size $K$ is 21 images (1%) of the training dataset. For CEAL and SSAL methods, the uncertain sample size, $K_u$, stands at 11, while the size of the pseudo label, $K_p$, is set to 5.

**Proposed method** Our SSAL method suggests selecting reliable pseudo-labels from the mid-range of uncertainty scores in the unlabelled pool, representing majority of samples, alongside expert labels of uncertain images to optimise annotation. A validation step is applied to choose the most confident pseudo-labels. Phases of our method are as follows:

- U-Net model is initially trained on $L$. At each AL iteration, uncertainty scores are computed for every image in $U$ using the model's prediction. Images in $U$ are ranked based on these scores, and the most uncertain samples, $K_u$, are queried for annotation.

- Uncertainty scores for remaining $U$ samples are normalised; The highest frequency bin in the uncertainty histogram determines the range of images as candidate pseudo-labels $C_p$.

- A threshold-shifting is applied to validate $C_p$ quality . In binary segmentation, a 0.5 threshold categorises pixels as foreground (1) or background (0), based on predicted probability. we analysed masks created at 0.4, 0.5, and 0.6 thresholds, focusing on variance. Low variance indicates strong model confidence, signifying stable segmentation across threshold adjustments. Reliable $K_p$, the size of pseudo-labels, is chosen from low variance predictions, with post-processing applied to $K_p$ prior to training.

- The batch, K, including $K_u$ and $K_p$, is transferred from $U$ to $L$, and the model is fine-tuned with the updated $L$. These steps are iterated until AL iterations are met.

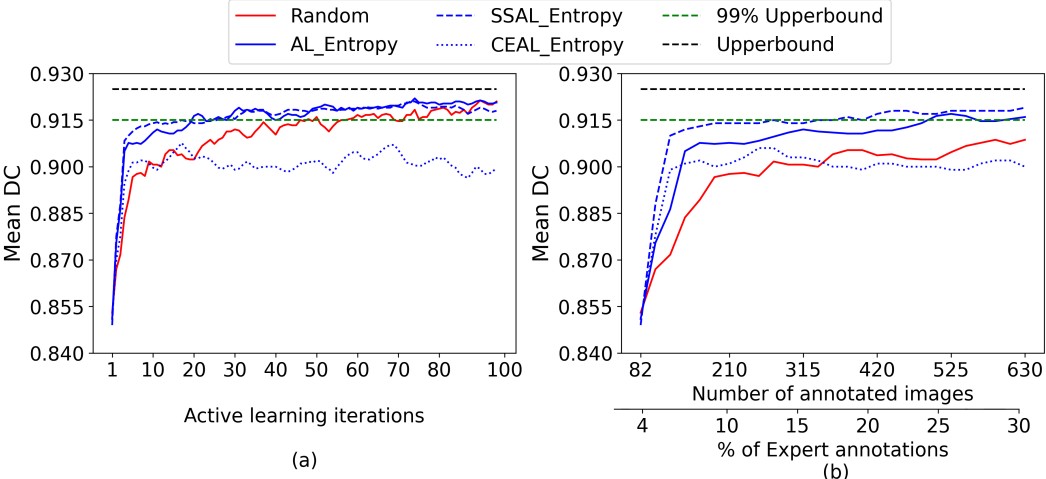

Figure 1: Performance profiles for Random, AL, CEAL, and SSAL (ours) using Entropy. Black and green dashed lines indicate the upper bound and 99% upper bound, respectively. (a) shows the performance in (Mean DC) at each AL iteration; (b) presents the performance of each method and the number of expert-labelled images utilised.

## 3. Results and Discussion

In Figure 1, our proposed SSAL_Entropy method surpassed Random, AL_Entropy, and CEAL_Entropy techniques, reaching 98% maximum accuracy with only 7% annotations, while Random required 30% labelled images for comparable performance.

Similarly, SSAL_Entropy achieved 99% accuracy with 10% annotations, compared to the 24% needed by AL_Entropy, indicating a 14% reduction in annotation cost (equivalent to 294 fewer labelled images). CEAL approach maintained performance around 97% of the upper bound throughout AL iterations. Table 1 displays our method's performance against the competitors.

Our method's efficiency is due to its ability to select pseudo-labels from a range of images representing most of the unlabelled set.

In conclusion, SSAL_Entropy outperformed competing strategies, achieving high segmentation accuracy with significantly fewer annotations, thus offering substantial labelling cost savings. Future work will enhance our method by integrating a verification network to evaluate the pseudo-label quality and exploring its generalisability to diverse medical imaging.

Table 1: Segmentation performance for SSAL_Entropy compared to various AL methods at selective percentages of annotations; performance is given as the ratio of upper bound.

| Percentage of labels | 0.06 | 0.08 | 0.10 | 0.15 | 0.20 | 0.25 |
|---|---|---|---|---|---|---|
| SSAL_Entropy | 97% | 98.7% | 99% | 99% | 99% | 99.3% |
| CEAL_Entropy | 96.7% | 97.1% | 97% | 97.5% | 97% | 97% |
| AL_Entropy | 96.5% | 98% | 98% | 98.7% | 98.5% | 99.3% |
| Random | 93.5% | 95.5% | 96.7% | 97% | 97.6% | 97.2% |

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
