# OpenReview forum: "Semi-supervised Active Learning for Left Ventricle Segmentation in Echocardiography"
_MIDL.io/2024/Short_Papers — MIDL 2024 Short Papers_

### Official Review · Reviewer_UbPA · 2024-04-24

**Confidence:** 2
**Final Rating:** 3.5

**Review:**

I am not an expert in active learning.

It seems that the authors start with a unet trained on 80 images, and are then proposing a heuristic for choosing images with uncertainties (based on pixel-wise confidence and thresholding). This method performs slightly (hard to tell) better than other random and entropy-based methods.

A main concern I have is that the training starts with 80 images -- at that point, it feels like care should be given to the unet training rather than active learning. My guess is that with proper (and substantial!) augmentation, the unet can do substantially better, but this is not discussed. I would propose that with that much initial data, the active learning should not be needed.

There are also substantial models now that can likely do the segmentation better, if interactions are, indeed, allowed, like SAM/MedSAM/Scribbleprompt, which can solve problems de novo with just clicks. They are not active learnign strategies, but can also perform quite well with interaction.

---

### Decision · Program_Chairs · 2024-04-26

Accept